# FEM-Analysis of 2D Micromachined Flow Transduers based on aGe-Thermistor Arrays and a Double Bridge Readout

**DOI:** 10.3390/s19163561

**Published:** 2019-08-15

**Authors:** Almir Talic, Samir Cerimovic, Roman Beigelbeck, Franz Kohl, Thilo Sauter, Franz Keplinger

**Affiliations:** 1Department for Integrated Sensor Systems, Danube University Krems, Viktor-Kaplan Straße 2, A-2700 Wiener Neustadt, Austria; 2Institute of Sensor and Actuator Systems, Vienna University of Technology, Gusshausstraße 27-29, A-1040 Vienna, Austria

**Keywords:** micro-electro-mechanical systems (MEMS), wind sensor, finite element method (FEM), Wheatstone bridge configuration

## Abstract

This paper reports on a design and simulation study aiming at high-accuracy 2D micromachined thermal flow transducers. The scope is restricted to micromachined devices featuring a square-shaped membrane incorporating central symmetric thin-film devices. A microthermistor array probed spatial excess temperature variations while the main heat supply was alternatively established by optional heating resistors or by pronounced self-heating of the thermistor devices. Proper device designs enable leading edge transducer performance without sophisticated signal conditioning schemes. We found that a high azimuthal uniformity of flow magnitude transduction is tantamount to a precise azimuthal accuracy. The most advanced result gave a maximum azimuthal aberration of 0.17 and 1.7 degrees for 1 m/s and 10 m/s, respectively, while the corresponding magnitude uniformity amounted to 0.07% and 0.5%. Such excellent specifications exceed the need of ordinary meteorological applications by far. However, they are essential for, e.g., precise non-contact measurements of 2D relative movements of two quasi-planar surfaces via the related Couette flow in intermediate air gaps. The simulations predicted significantly better device characteristics than achieved by us in first experiments. However, this gap could be attributed to imperfect control of the flow velocity field by the measurement setup.

## 1. Introduction

After more than 40 years of research, miniaturized thermal flow transducers based on thermopiles as well as on PTC or NTC (positive or negative temperature coefficient of resistivity) thermistors are mature for crucial applications in, e.g., industrial process control [1,2,3,4,5,6,7,8]. Micro-machined thermal flow transducers are favorable for spot measurements of the flow velocity field since, in general, shrinking their dimensions improves sensitivities and response time on the one hand and reduces power requirements on the other hand. This work deals with design aspects of membrane-based, micro-machined 2D calorimetric flow transducers that sense magnitude and direction of fluid flow velocities in close vicinity to their membrane surfaces. Calorimetric transduction typically comprises several transduction steps: (i) Electrical power is dissipated as heat by means of a heating resistor; (ii) conjugated heat transfer takes place in case of a passing fluid, leading to a modification of the temperature distribution within the transducer membrane; (iii) various thermoelectric or resistive transducers can be incorporated in the membrane which convert local excess temperatures into electrical quantities, i.e., currents, voltages, or resistances, for example; and (iv) electronic circuits convert these intermediate quantities into analogue readings or digital data to complete sensor functionality.

Bidirectional calorimetric flow transducers typically comprise a heat source and temperature transducers located up- and downstream from this heater [8]. Any fluid volume passing by the heater acquires thermal energy and delivers a certain amount by convection to the downstream membrane region. Therefore, convective heat transfer generates temperature differences between up- and downstream membrane locations that are sensed by the temperature transducers and further related to the fluid motion and properties.

Micromachined flow transducers often comprise a frame machined from Si wafers that clamps a thin multi-layer membrane incorporating thin-film devices for heating and temperature sensing. Temperature transducers in form of microthermistors made of thin-film amorphous germanium (aGe) are best suited for demanding flow sensing applications [9,10,11]. These devices offer high temperature resolution, small heat capacity, and fast response. An appropriate specific resistance of the NTC thermistor material enables both low spurious self-heating with moderate excess temperatures and, if desired, forced self-heating for calorimetric as well as anemometric flow transduction without a dedicated heating element [12]. Simulation studies have shown that for a given micro-machining technology, designs can be optimized regarding high initial flow sensitivity, wide measuring range, or high response speed [10].

Pronounced bidirectional flow transducers have been developed based on such thermistors for flow rate measurements in tubes [9]. They usually exhibit approximate sinusoidal dependences on the azimuth of the flow velocity direction with respect to the alignment of the thin-film elements on the sensor membrane. An orthogonally arranged pair of such flow transducers have successfully been used to track the y/x position of a computer mouse [13,14] Embedded in the sliding plane of the mouse, they simply sense the Couette flow of air in the gap between the moving mouse and the mouse pad surface with respect to the mouse movement.

In a straightforward attempt, two of such bidirectional transducers could be packed on a single membrane in order to separately sense two orthogonal components of a flow velocity simultaneously with a single 2D flow transducer. However, the complex interaction of these two thermal subsystems have to be considered for a satisfying result. Consequently, numerous attempts have been made to establish appropriate 2D flow transducers [15,16,17,18,19,20,21,22,23,24,25,26,27,28,29], which are often referred to as wind sensors. The vast diversity of investigated transducer implementations and related requirements on signal conditioning strategies can be perceived from the summary [25]. Even for a given device technology, many obstacles have to be overcome in order to design a miniaturized 2D flow transducer that combines high angular accuracy and uniform magnitude transduction over the full 360 degree range with good magnitude sensitivity and a wide magnitude range. Based on a novel design, we will present simulation results on these key characteristics and compare them with preliminary experimental results.

## 2. Materials and Methods

The models simulated for this study closely approximate device implementations obtained with Si-micromachining and a mature technology for aGe thermistors. The details of the device production are described in [30]. Therefore, the obtained simulation results are strictly valid only for comparable transducer technologies. In this paper, however, we focus on design-related impacts on the flow transduction mechanism. Therefore, the presented conclusions contain useful hints for many other technological approaches to micromachined flow transducers.

### 2.1. Design Considerations

On the way from simple bidirectional thermal flow sensors towards wind sensors, a couple of decisions at system level and design considerations are mandatory in order to achieve superior device performance. Only a restricted part of the vast design possibilities for such devices can be covered by a single research paper. Consequently, we describe only our investigations on the most successful approaches. These 2D flow transducers have square membranes containing eight semiconductor thermistors and optional thin-film heating resistors.

Readout architectures based on resistance bridges are a common choice in almost any case of thermal flow transduction. Four matching membrane thermistors enable an expedient readout of small asymmetric temperature variations within the membrane. To boost the thermal actuation by self-heated thermistor bridges, the setups can be complemented with dedicated heating resistors. For a given bridge supply voltage $U_{S},$ full bridges offer twice the output of half bridges composed of a single upstream and downstream thermistor combined with two external resistors. Furthermore, any common scaling of the four thermistor resistances, e.g., by changes of the ambient temperature or technology variations, has no perceptible effect on the offset voltage and the conversion of the bridge supply voltage into a bridge detuning signal. Finally, a readout based on a perfect bridge yields zero output signal for zero flow.

As we will show below, there is another important benefit of bridge-based readouts that applies exclusively to 2D flow transducers. When compared with the azimuthal dependence of thermistor temperatures, bridge-based flow transduction signals facilitate a much better approximation to the desired sinusoidal characteristics.

The layout of Figure 1a arises immediately from a simple bidirectional flow sensor characterized by single thermistors situated up- and downstream of a central heating resistor [14] by splitting each of them into two equal parts. Interdigitated contacts to the thermistor film deal with the high resistivity thermistor material. Four thermistors enable a Wheatstone bridge configuration for convenient readout of temperature differences between the upper and lower, or, alternatively, left and right half of the device.

A pronounced sensitivity of the bridge voltage to flow in parallel to the y coordinate can be expected if the upper thermistor pair $R_{A},R_{B}$ forms one bridge diagonal while the lower pair $R_{C},R_{D}$ constitutes the other one. Temperature variations in the  x direction are suppressed effectively by this configuration. The numerator of the Wheatstone formulas in Equation (1) governs the magnitude of the related bridge unbalance voltages $U_{\mathrm{By}}$,
(1)$$U_{\mathrm{By}}=U_{S}\cdot \frac{R_{C}R_{D}-R_{A}R_{B}}{\left(R_{A}+R_{D}\right)\cdot \left(R_{B}+R_{C}\right)}\approx U_{S}\cdot \frac{R_{C}R_{D}-R_{A}R_{B}}{\left(R_{A}+R_{C}\right)\cdot \left(R_{B}+R_{D}\right)}.$$

The different denominators, resulting from the interchange of the two resistors of a single bridge diagonal, causes only a very moderate change of $U_{By}$, as the four resistance values are approximately equal. $U_{S}$ denotes the bridge supply voltage.

For flow along the x coordinate, the bridge thermistors are laterally positioned with respect to the resistive heater. As the preferred directions of convective and conducive heat transfer are virtually orthogonal, this configuration is suspected to yield smaller temperature differences due to convection. The best transduction of the *x*-component of the velocity can be achieved by swapping the position of *R*_A_ and *R*_D_ in Figure 1b.

(2)$$U_{\mathrm{Bx}}=U_{S}\cdot \frac{R_{A}R_{C}-R_{B}R_{D}}{\left(R_{C}+R_{B}\right)\cdot \left(R_{A}+R_{D}\right)}\approx U_{S}\cdot \frac{R_{A}R_{C}-R_{B}R_{D}}{\left(R_{C}+R_{D}\right)\cdot \left(R_{A}+R_{B}\right)}.$$

If both resistance bridge configurations are utilized alternately, two linearly independent flow signals are available. Therefore, 2D flow reconstruction is feasible with this membrane design. A matrix of switches is required for reconfiguration of the resistance network but simultaneous reading of both signals, as mandatory for fast azimuth measurements, is not feasible.

However, devices according to Figure 1 provide an appropriate starting point for the development of 2D flow sensors by thorough simulation studies of flow magnitude and direction transduction. The simulated polar diagrams of Figure 2 illustrate the azimuth dependencies of $\left|U_{\mathrm{Bx}}\left|,\right|U_{\mathrm{By}}\right|$ for a certain flow rate of a device according to Figure 1a. Circular characteristics in the upper as well as in the lower hemisphere indicate a characteristic proportional to |sinφ| while a |cosφ| dependency results in circles in the left and right hemisphere of Figure 2. A close inspection of Figure 2 reveals minor deviations from the ideal circular shapes. The conversion signal $\left|U_{\mathrm{By}}\right|$ saturates slightly around its maximum, while the opposite phenomenon is observed for the $\left|U_{\mathrm{Bx}}\right|$ characteristic.

A sinusoidal dependency on the azimuth angle indicates, e.g., that the y velocity component (see Figure 1a) governs the azimuthal variation of the main transducer output. The second double-lobe confirms that the alternative bridge configuration may be used to record the x velocity component via $U_{Bx}$. The signal diagrams of Figure 2 can be approximately modeled as
(3)$$\left|U_{Bx}\right|\;\propto \;f_{x}\left(\left|v\right|\right)\cdot \left|\cos\left(\varphi \right)\right|,\;\left|U_{By}\right|\;\propto \;f_{y}\left(\left|v\right|\right)\cdot \left|\sin\left(\varphi \right)\right|$$
where v denotes the average flow velocity in the channel and φ the azimuth angle between the flow direction and the x axis. The magnitude dependencies $f_{x}\left(\left|v\right|\right)$ and $f_{y}\left(\left|v\right|\right)$ must be obtained from calibration experiments. In spite of the directional dependencies shown in Figure 2, the transducer design according to Figure 1a certainly does not provide means for continuous recording of flow direction and magnitude. The required periodic interchange of bridge resistors prohibits continuous flow velocity transduction. Using two orthogonally oriented bridge transducers of the shown kind, one could continuously resolve flow direction φ and magnitude |v|, although with a moderate spatial resolution simply from
(4)$$\tan\left(\varphi \right)=U_{B2}/U_{B1},\;\mathrm{and}\;\left|v\right|=f\left(\sqrt{U_{B1}^{2}+U_{B2}^{2}}\right)$$
where the indices 1 and 2 distinguish the devices and $U_{B}$ is the bridge signal for the preferred orientation of each device (i.e., $U_{By}$ in Figure 2). Function *f* denotes the magnitude dependence for the preferred orientation (i.e., *f*_y_ in Equation (3)).

It would be mandatory for highest spatial resolution and beneficial for economic manufacturing, if the two separated transducers could be merged into a more sophisticated combined transducer design that fits to a single MEMS membrane. Hence, eight membrane thermistors, forming two permanent resistance bridges, must be suitably arranged aside with proper designed heating elements. However, any mutual interference between the orthogonal subsystems must be considered to ensure proper wind sensor operation.

Numerous unique flow transducer implementations have been published in the past [1,2,3,4,5,6,7,8,9,10,11,12,13,14,15,16,17,18,19,20,21,22,23,24,25,26,27,28,29,30,31] but reports on comprehensive exploration of the design space of a given technological approach are not available yet. We have investigated a total of twelve basic flow transducer designs, some of them including subsets, but all simulation models adhere to the aGe microthermistor technology. Each promising design was explored for optional bridge configurations and alternative operational modes with respect to their flow transduction characteristics. The key parameters of 2D-transducer performance are azimuthal uniformity of the flow magnitude conversion as well as the deviation between the flow direction and the estimated azimuth angle aberration. Due to the limited space, only the best performing transducer design is extensively discussed in this paper. Other designs and results are shown in Appendix A to this paper (Table A1).

### 2.2. FEM Modeling

We conducted comprehensive FEM studies of the conjugated heat transfer on a number of promising thermistor array concepts aiming at proper arrangements of heating resistors as well as temperature sensing elements on transducer membranes. Several designs were identified that enable efficient and accurate conversion of flow magnitudes and directions. A 3D FEM modeling is required because of the 2D extension of the membrane elements and an imposed flow velocity profile in the third direction. A modeling approach will be shortly explained by means of the FEM model of the calorimetric transducer depicted in Figure 1a. FEM models of all other sensors presented in this paper are similar and differ only in dimensions of membrane and active elements (i.e., heaters and thermistors). Figure 3a depicts a schematic cross section in the *y*-direction through the thermistors *R*_C_ and *R*_A_ (see Figure 1a). The membrane of this particular transducer has an area of 0.5 × 1 mm², whereas the membranes of all other sensor layouts presented in this paper measure 1.2 × 1.2 mm². Each membrane is suspended over a 350 μm thick silicon frame and consists of a SiO_2_, Si_3_N_4_, and SiN_x_ layer sandwich with an overall thickness of 1.57 µm. For simulations, all three layers are combined to one single layer using averaged thermal properties. The temperature sensing elements are embedded 0.32 µm above the bottom membrane surface. A single thermistor measures 400 × 35 μm², exhibiting a total thickness of 0.27 µm. The interdigital metal layers used to contact the thermistors (see Figure 1a) were not considered by this simple 3D modeling approach. The U-shaped heater consists of two stripes, each 5 µm wide and about 1.2 mm long, with a 15 µm spacing between them. For the sake of simplicity, it was simulated as a single block featuring the same thickness as the adjacent thermistors (Figure 3b). Due to the high aspect ratio of the membrane and the embedded components, the number of required mesh elements is very high. However, it can be significantly reduced by scaling these subdomains with a factor a = 20. In order to obtain approximately the same temperature distribution, the material properties must be appropriately scaled, too. For more details about the FEM modeling refer to [28].

A rectangular flow channel of 1mm height and 12 mm width was assumed in accordance with the dimensions of the rectangular flow channel used for experiments. Thus, the modeled flow velocity field exhibits no variation in flow direction and a parabolic dependence across the orthogonal plane to maintain non-slip boundary conditions at the top and bottom boundary of the flow channel. The trapezoidal cavity below the sensor membrane is filled with resting fluid. Its shape accounts for the anisotropic KOH-etching step during the membrane fabrication.

Due to the imposed flow velocity, the solution of the general Navier–Stokes equation is not necessary and only the heat transfer equation incorporating conductive and convective heat transfer must be solved. The boundary condition at the in- and outlet of the flow compartment is implemented as convective flux, the remaining parts of the model circumference were kept at the ambient temperature. This is also the initial temperature condition for all domains.

The power density of the heaters is specified while the heat dissipation of the bridge thermistors is computed from the actual thermistor temperatures in conjunction with the applied bridge supply voltage or current levels. In case of self-heating thermistor operation, an iterative solution of the model is scheduled.

Simulations were performed by means of the commercial finite element analysis software COMSOL Multiphysic^®^. Figure 4 shows membrane regions of three high-performance transducer designs. The thermistors feature triangle (a) or trapezoidal (b) shape. A circular arrangement of eight rectangular thermistors (c) was also considered. Moreover, two operational modes for each transducer were considered. In the calorimetric mode, the heaters were used as a heat source (Figure 5). Eight membrane thermistors were appropriately connected to form two Wheatstone bridges, each supplied with the same constant voltage. As the supply voltage is assumed to be very low ($U_{S}\;$= 1V), the self-heating effect of the thermistors can be neglected in this operating mode.

In the second operating mode (Figure 6), the heaters were switched off and the self-heating effect of the thermistors solely was utilized as a heat source. The bridges were connected in series and supplied with a constant current, rather than with a constant voltage, as it is the case in calorimetric mode. If a thermistor temperature suddenly increases, its resistance decreases due to the negative temperature coefficient (NTC, α = −0.02 K^−1^ [9]). In case of a voltage supply, this results in a rise of the electrical current through the thermistor and a further increase of its temperature. This positive feedback can lead to the thermal destruction of the thermistors. Thus, in the self-heating mode, the constant current supply should be preferred.

Table 1 gives an overview of the best performing transducer layouts and their respective operational modes. The flow transduction errors of three high-performance design alternatives (triangle, circular, trapezoidal) are given for 1 and 10 m/s, respectively. All twelve layouts of the transducer membrane region investigated in the course of this research are unveiled in the supplementary matter. Results obtained for self-heating operation of membrane thermistors are listed along with two cases of the calorimetric operating mode.

According to Table 1, best performance is obtained by the “triangular” design in conjunction with the calorimetric mode. The related membrane layout is shown in Figure 7. The rest of this paper is devoted to describe the underlying synergy of diverse influences as unveiled by the FEM simulations.

### 2.3. Best Performing Membrane Design

Eight thermistors were “triangularly” arranged together with four slim heating resistors on the membrane. The large triangular thermistor shapes are suspected to promote a large range of efficient transduction of flow magnitudes. The four straight heaters, connected in parallel or series, ensured a four-fold rotational symmetry of power dissipation in the membrane.

Figure 8 depicts simulation results for the eight thermistor resistances as a function of the flow direction. As we deal with NTC thermistors, resistance maxima correspond to temperature minima and vice versa. Because of the small relative changes, the thermistor temperature curves look like vertically mirrored resistance curves. The shape of these curves is typical for the impact of the convective heat transfer. Around the upstream position (maximum resistance) of a specific thermistor, there is a broad azimuth region where convective cooling dominates resulting in elevated resistance values. In contrast, lowering of resistances takes place only within a smaller azimuthal region around the downstream position. Downstream thermistors are less cooled or even heated by convection since the fluid temperature is somewhat elevated within a narrow lobe in the wake of the power dissipating elements in the membrane. Thus, the different curvature around the maxima and minima of these resistance characteristics is a common feature of all 2D calorimetric flow transducers.

With respect to increments of the azimuthal coordinate φ, the rate of resistance decrease of each characteristic of Figure 8 differs from the rate of its increase, if both are taken at a certain resistance level. The skewed form of the resistance characteristics is a consequence of the triangular shape of each thermistor that is not symmetric with respect to the azimuthal coordinate φ (see Figure 7). The continuous traces (thermistors 1, 3, 5, 7) are skewed left whereas the dashed curves (thermistors 2, 4, 6, 8) are skewed right. As will be shown below, the different skewness of neighboring thermistors enables a better approach to the desired sinusoidal characteristics of double bridge flow transduction.

The convection induced membrane temperature change leading to the characteristics of Figure 8 is depicted in the right sub-figure of Figure 9 for a flow direction parallel to the *x*-axis. The shown contour maps cover the transducer membrane measuring 1.2 × 1.2 mm. The applied temperature units are in Kelvin while the diagrams are computed for a total heating power of 3 mW and a flow velocity of 1 m/s. As expected, convective cool down is strongest slightly upstream of the membrane center while a moderate temperature elevation will eventually appear in the downstream region, which is typical for moderate flow velocities. In contrast to the smooth characteristic of convection-induced temperature change, the temperature distribution itself, as depicted in the left sub-figure of Figure 9, exhibits strong local variations throughout the membrane.

### 2.4. Wheatstone Bridge Readout Options

The eight thermistors of the design depicted in Figure 7 (confer also Figure 5a, where the thermistors are denoted as *R*_1_–*R*_8_) enable four distinct connection schemes for orthogonal pairs of Wheatstone bridges.

The Table 2 notation for the “wide” bridge configuration and pronounced $\pm v_{x}$ sensitivity, for example, translates into
(5a)$$U_{\mathrm{Bx},\mathrm{wide}}=U_{\mathrm{supply}}\cdot \frac{R_{\mathrm{TH}2}R_{\mathrm{TH}3}\;-\;R_{\mathrm{TH}6}R_{\mathrm{TH}7}}{(R_{\mathrm{TH}2}\;+\;R_{\mathrm{TH}6})\left(R_{\mathrm{TH}3}+R_{\mathrm{TH}7}\right)}$$
for the bridge detuning voltage.

An alternative operational mode that is exclusively based on the self-heating operation of the membrane thermistors supplies a controlled current $I_{\mathrm{supply}}$ to the thermistor bridges. In this case the equivalent to Equation (5a) reads
(5b)$$U_{\mathrm{Bx},\mathrm{wide}}=I_{\mathrm{supply}}\cdot \frac{R_{\mathrm{TH}2}R_{\mathrm{TH}3}-R_{\mathrm{TH}6}R_{\mathrm{TH}7}}{R_{\mathrm{TH}2}+R_{\mathrm{TH}6}+R_{\mathrm{TH}3}+R_{\mathrm{TH}7}}$$

Depending on the simulated characteristics of thermistor resistances versus flow direction depicted in Figure 8, directional diagrams of the flow transduction can be computed for all four bridge alternatives. A closer inspection of eight distinct variants of Equation (5a) reveals that at least for small variations of the thermistor temperatures the denominator has negligible influence on the directional characteristic whereas the numerator constitutes a sensitive measure of any resistance variation that is asymmetric with respect to the membrane center.

Note that in all numerators according to the scheme of Equations (5a) and (5b), only products of a left-skewing with right-skewing resistance characteristics (see Figure 8) occur. Hence these products are no more skewed with respect to the azimuthal coordinate. The highest dynamic range of a single product can be expected, if the resistance factors are composed from neighboring thermistors on the membrane, i.e., for the “wide” and the “diagonal wide” cases. Furthermore, multiplication of neighboring resistance characteristics promotes a broadening around the product minima and enhance the curvature around the product maxima. Thanks to the thermistor shapes, the bridge detuning signals exhibit much better approximation to sinusoidal azimuth dependencies compared to the azimuthal resistance characteristics of individual membrane thermistors. In fact, this consequence of the Wheatstone bridge readout technique is a mandatory prerequisite to obtain accurate flow vectors by immediate evaluation of the analytical functions given by Equation (4).

Figure 10 depicts directional diagrams of the flow transduction signals generated from simulated bridge detuning signals of four distinct combinations of two perpendicular Wheatstone bridges. They reveal the strong dependency of the output signals on the involved combination of thermistor locations on the membrane. The azimuth angles of the diagrams refer to the flow velocity azimuth *φ* where *φ* = 0 coincides with the *x*-axis of Figure 7. To highlight the influence of the velocity magnitude, directional diagrams for 1 m/s as well as 10 m/s are illustrated in Figure 10.

Owing to the concurrence of preferential direction and largest distance to the Si frame, the most efficient flow transduction is obtained by the “diagonal wide” configuration. The slightly less sensitive “wide” configuration excels with the most uniform conversion of the flow magnitude with respect to the azimuth angle. In contrast to intuition, the magnitude of the voltage vector for the “narrow” configuration is largest for odd multiples of π/4. In this case, constructive contributions by the action of both heater orientations occur, which boosts the flow conversion efficiency. The most intriguing directional characteristic, however, is obtained for the “diagonal narrow” configuration, where occasionally flow direction and voltage vector show opposite azimuthal progress (see Figure 10b). Obviously, a destructive superposition of contributions of the two orthogonal heater branches occurs.

All simulated flow transduction signals are sub-linear functions of the mean fluid velocity in the flow channel of the model. However, the saturation of flow transduction is most pronounced for the “diagonal narrow” and the “wide” bridge configuration as revealed by the comparison of Figure 10a,b.

Only the “wide” configuration of the Wheatstone bridge delivers a directional characteristic that promises a uniform transduction of the flow velocity magnitude without the need for extensive post processing of the magnitude output. The following results refer always to this configuration. 

Figure 11 presents the uniformity of the computed magnitude conversion as a function of the flow direction. The relative signal magnitude of this configuration is within an interval of 1 ± 0.006 for all flow angles and magnitudes. Remarkably, the magnitude variation for 1 m/s is smaller than that of 0.1 m/s. In case of the smallest flow rate, the magnitude variation closely approximates a −cos(4*φ*) dependence. All traces contain such variations with a period of π/2 or 90 degrees. This is a consequence of the period of π for each Wheatstone bridge arrangement and the π/2 rotation between the two orthogonally oriented bridges employed for flow magnitude and azimuth transduction. Although the quantitative effect of strong convection on the transduction uniformity seems very moderate, the qualitative dependence on the azimuth angle becomes quite different due to the emergence of higher harmonics. This behavior demonstrates the complex interdependences between design and strong convective heat transfer. At the highest flow rate, the strongest variation of magnitude conversion is seen for flow directions close to the ±*x* and ±*y* axes.

The markers on the curves in Figure 10 indicate uniform increments of the rotation angle of the flow direction by five degree each. The stronger the flexion of the directional characteristic, the tighter the marker positions. Therefore, the direction of the 2D vectors composed of the detuning signals of orthogonal thermistors bridges do not coincide perfectly with the respective flow direction. Figure 12 highlights the azimuthal deviation between the actual flow direction and the azimuth angle computed from simulated transducer signals for the preferable “wide” bridge configuration. The diagram reveals that the aberration changes its sign between 0.1 m/s and 1 m/s. Beyond 1 m/s the aberration grows drastically when the flow velocity is increased. 

The azimuth deviation is largest for the highest flow rate with peak values of about ±1.5 degree located about ±15 degrees from the coordinate axes. All traces show a variation period of π/2. The trace for a mean flow velocity of 0.1 m/s is the smallest and approximates a sin(4*φ*) azimuthal dependence very well.

Just for reference, the computed peak-to-peak azimuth and magnitude deviation of the “narrow” configuration of Wheatstone bridges amounted to 12 degrees and 20%, respectively, at a flow velocity of 1 m/s.

Figure 13 shows simulated bridge detuning voltage *U_Bx_* for three heater configurations, i.e., (i) all four heating resistors operated (total), (ii) the two heaters extended in *x* direction are heated exclusively (only_EW_on), and (iii) only the heaters extended in *y* direction are operated (only_NS_on). Each characteristic is displayed together with respective fits with cosine functions. The flow magnitude conversion induced by a single pair of heating resistors exhibits marked deviations from the wanted sinusoidal azimuth dependence, as Figure 13 illustrates. However, the different partial heater operations cause opposite signal deviations from ideal cosine characteristics. As these contributions are superimposed to the total signal, a nearly perfect compensation of conversion errors takes place. Thus, the excellent uniformity of the flow magnitude conversion envisioned in Figure 11 turned out to be a beneficial coincidence. Hence the action of all four heating resistors is required to achieve bridge detuning signals that deliver a nearly perfect magnitude uniformity over the full azimuthal range.

The nearly perfect sinusoidal azimuthal dependency of the total $U_{\mathrm{Bx}}$ signal together with the π/2 rotational symmetry of the membrane layout enables the evaluation of the transducer signals from the expressions
(6)$$\left|v\right|=f\left(\sqrt{U_{\mathrm{Bx}}^{2}+U_{\mathrm{By}}^{2}}\right)\;\mathrm{and}\;\varphi =\arctan\left(U_{\mathrm{By}}/U_{\mathrm{Bx}}\right)$$

## 3. Measurements

The sensor chip was mounted flush with the bottom of a rectangular flow channel of 12 mm width and 1 mm height. The channel axes can be rotated 360° around the normal to the center of the membrane surface. Filtered nitrogen was used for the gas flow. The velocity of the nitrogen in the channel was adjusted by a mass flow controller with a range up to 2000 cm^3^/min, which corresponds to a maximum velocity of 27 m/s. The flow guidance setup has been described in detail in [20,21]. Figure 14 depicts a diagram of *U*_By_ vs. *U*_Bx_ measured with a transducer featuring the design of Figure 7, where the zero-flow bridge offsets have already been subtracted. The intended position of the transducer surface is flush with the bottom wall of a rectangular flow channel.

The circle corresponds to a perfect direction independent magnitude conversion. The observed magnitude variation as well as the corresponding azimuth deviations at a cross-sectional mean flow velocity of 1 m/s, visible in Figure 14, were much larger than the corresponding simulation results. These experimental deviations resulted mainly from bumps situated at two opposite edges of the transducer chip. These bumps consisted of epoxy resin and were protecting vulnerable bonding wires (Figure 15a). They constricted the regular cross-section of the flow channel (Figure 15b), causing azimuth aberration and magnitude irregularities of the flow velocity field near the transducer membrane surface: Imperfections of manufacturing processes generating, e.g., unmatched thermistor resistances have much less impact on the transducer performance.

In agreement with the bump positioning, both error characteristics of Figure 16 suggest a major variation with an azimuthal repetition of roughly 180 degrees. Around 0 and 180 degrees, where the bumps are in line with the flow channel axis, the measured flow magnitudes are smallest (negative relative error) while the smoothest region of this characteristic is found around azimuth angles of 90 and 270 degrees.

In view of the large deviations between measured and simulated characteristics, the measurements must be considered as a first attempt. A much-improved performance could be expected from transducer implementations featuring through-wafer vias and interconnection-lines at the backside of the chip.

## 4. Conclusions

Several direction resolving flow transducers featuring square-shaped micromachined membranes that incorporate eight thin-film thermistors together with optional heating resistors were studied by FEM simulations. The simulations were restricted to laminar flow tangential to the chip surface. 2D flow magnitude conversion and flow direction recognition becomes feasible with two Wheatstone bridges which are constituted by four membrane thermistors each. The related thermistor quartets feature a spatial offset by an azimuth angle of π/2. The investigated transducers family vary in the geometrical shape of the resistive elements, the transduction mode, and the interconnection schemes for electrical readout of the membrane thermistors. These variants yielded more than 60 different simulation models. However, only a couple of membrane layouts together with appropriate interconnection schemes gave highly satisfying transducer characteristics.

The FEM simulation studies revealed several decisive factors influencing the flow transduction performance. A minor directional dependency of flow magnitude conversion goes hand in hand with small aberration of the flow angle transduction. In order to enable a uniform flow magnitude conversion, a sinusoidal dependency of each bridge output signal on the azimuth angle is required. Two neighboring thermistors should form one diagonal of the associated resistance bridge while the opposite thermistors provide the other diagonal to achieve this goal. Each thermistor pair of a bridge diagonal should cover an azimuthal range of about π/4. Sufficient radial extent of thermistors gives a wide magnitude measurement range. In contrast, a significant azimuth offset between the thermistors of a bridge diagonal results in a much worse directional uniformity and huge aberration.

Good flow sensitivity requires sufficient upstream/downstream distance between heating resistors and the involves bridge thermistors while closer spacing extends the measuring range. For best conversion of the *x* component of the flow, the well separated thermistors 2, 3, 6, and 7 in Figure 7 are mainly responsible in conjunction with the heating resistors aligned along the y direction, i.e., the “wide” configuration. The action of the other two heating resistors enhances the flow conversion and improves the approximation to the desired sinusoidal variation of the bridge detuning signals as a function of the flow direction. The design example discussed herein offers in conjunction with the appropriate Wheatstone bridge configuration a nearly perfect azimuthal precision of flow velocity transduction over a wide flow magnitude range.

The high quality of the simulation results was hardly confirmable with practical flow measurement setups. The reason were strong irregularities around the surface of the 2D flow sensor that distorted the measurements. Therefore, the contacts to the membrane embedded thin-film devices have to be established from the backside of the silicon chip. Furthermore, trenches around the chip between transducer and PCB carrier have to be carefully filled up to guarantee a perfectly flat surface of the mounting. In the case of wind measurements, the mounting of the transducer should have perfect rotational symmetry. However, any fixation of the chip and interconnecting cables will interfere with the 3D flow velocity field to be measured, even when they are arranged at the rear side of the transducer chip. For these applications, the complete transducer assembly must be characterized by calibration experiments.

## Figures and Tables

**Figure 1 sensors-19-03561-f001:**
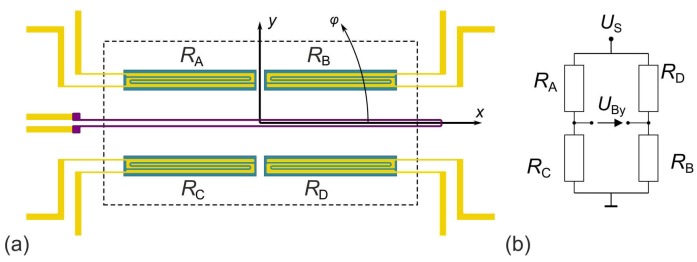
Membrane region of a simple extension of the calorimetric flow transducer layout. A slim heating resistor (purple trace) separates the rectangular membrane area (dashed rectangle) into upper and a lower region, each bearing a pair of membrane thermistors made of aGe (dark blue rectangles) with interdigitated contacts (yellow regions). This particular design (**a**) and Wheatstone bridge configuration (**b**) aims at pronounced sensitivity for flows along the *y*-coordinate.

**Figure 2 sensors-19-03561-f002:**
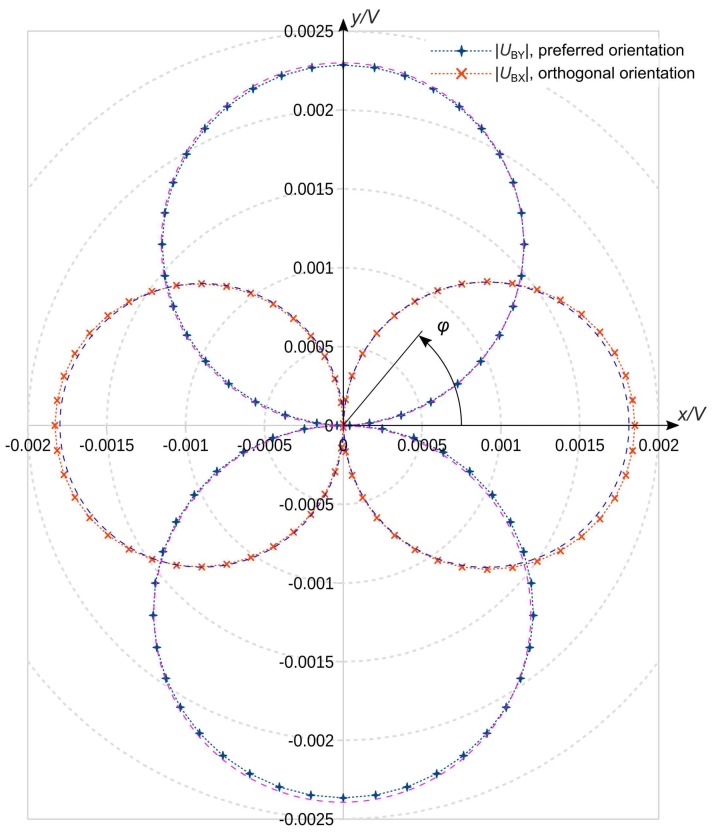
Simulated polar plots of the magnitude of two bridge unbalance signals versus flow azimuth angle for the design of Figure 1a. The tilt azimuth φ was manually incremented from 0° to 360° in uniform steps of 5°, the bridge supply voltage amounts to 1 V, the heating power to 2.5 mW and the NTC to 0.02 K^−1^. Dashed circles indicate the ideal |sin(φ)| and |cos(φ)| characteristics.

**Figure 3 sensors-19-03561-f003:**
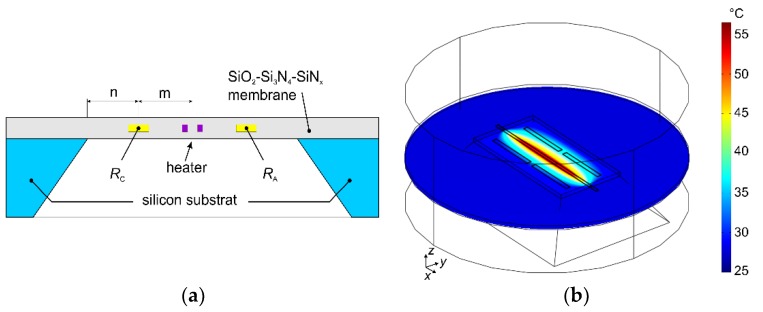
(**a**) Schematic cross section in *y*-direction of the flow sensor depicted in Figure 1a (m = 137.5 µm, n = 125 µm). (**b**) 3D FEM model of the flow sensor. It considers only a narrow cylindrical section around the sensor membrane and the flow channel above. The colors illustrate the temperature distribution at the cross section through a membrane and parallel to the *x–y*-plane. For this particular simulation a rectangular flow channel of 0.5 mm height and 1 mm width was assumed, with an airflow in *y*-direction and 1 m/s mean flow velocity. The heating power amounts to 3 mW and the ambient temperature to 25 °C (which is also the fluid temperature).

**Figure 4 sensors-19-03561-f004:**
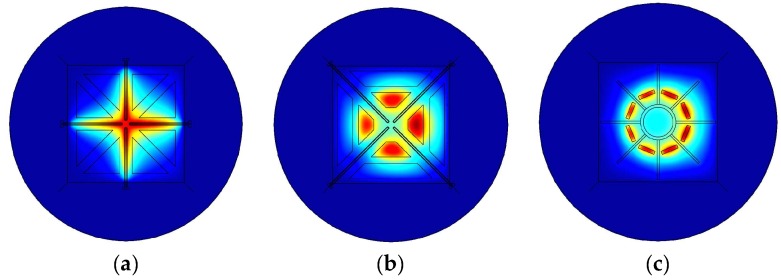
Membrane view (*x–y*-plane) of excess temperature of three design alternatives. All devices comprise eight thermistors (two orthogonal arrangements, each comprising four thermistors) and heaters. As a heat source serve either heaters (i.e., calorimetric operating mode as depicted in (**a**)) or a self-heating effect of the thermistors (**b**,**c**).

**Figure 5 sensors-19-03561-f005:**
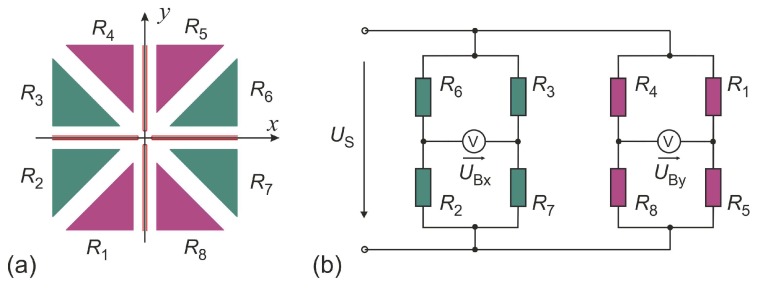
(**a**) Schematic illustration of the transducer with triangular shaped thermistors arranged as two orthogonal structures (each containing four thermistors). (**b**) The thermistors form two Wheatstone bridges supplied with a constant supply voltage $U_{S}$. Heaters (red colored) serves as a heat source.

**Figure 6 sensors-19-03561-f006:**
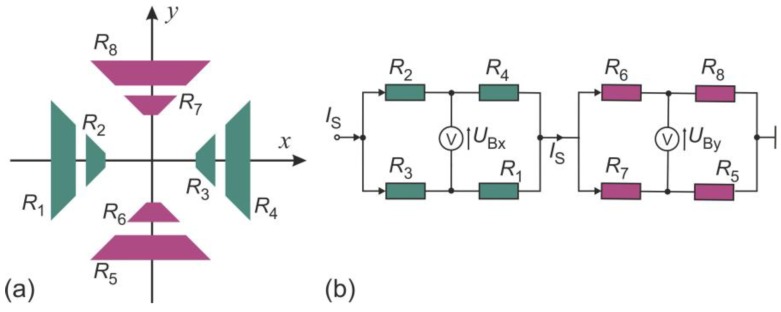
(**a**) Schematic illustration of the transducer with trapezoidal shaped thermistors arranged as two orthogonal structures (each containing four thermistors). (**b**) The thermistors form two Wheatstone bridges supplied with a constant current *I*_S_. Self-heating effect of the thermistor serves as a heat source.

**Figure 7 sensors-19-03561-f007:**
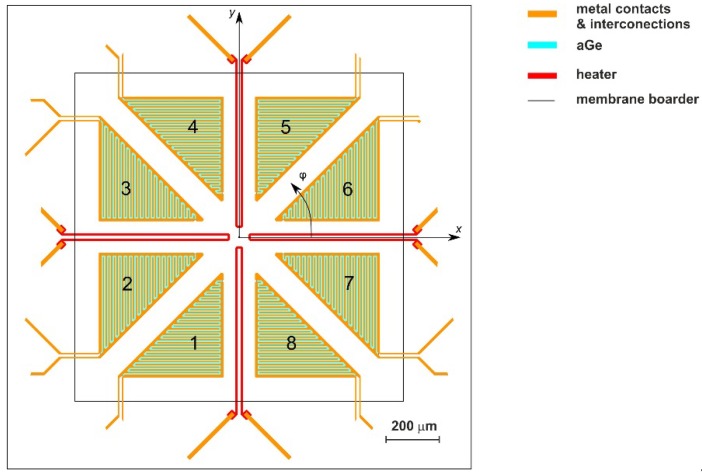
Best performing membrane design comprising eight triangular thermistors arranged side by side with four slim heating resistors on a square membrane. The four linear heaters are connected in parallel or in series to ensure a fourfold rotational symmetry of power dissipation in the membrane. The large thermistor areas are beneficial for uniform transduction of flow magnitude and flow direction. Thermistor numbering refers to the characteristics of Figure 8.

**Figure 8 sensors-19-03561-f008:**
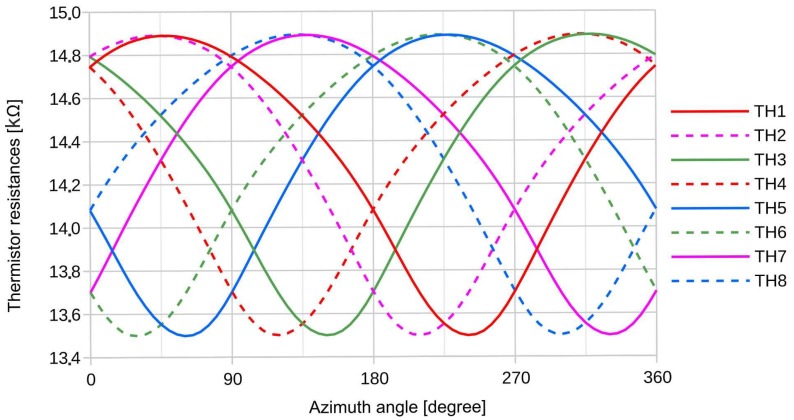
Simulated resistance variations with the direction of flow. The relative variation due to a 360° change of the flow direction of the thermistor resistances is of the order of 10% reflecting a temperature variation of ~5 K.

**Figure 9 sensors-19-03561-f009:**
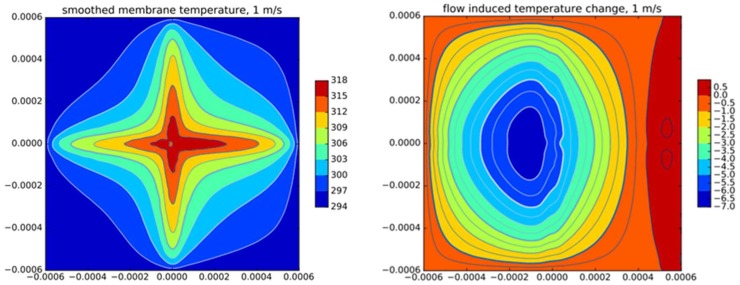
(**Left**): Contour plot of the membrane temperature field for a flow rate of 1 m/s from left to right. (**Right**): Corresponding contour plot of the convection induced temperature change in Kelvin at the surface of transducer membrane. A total heating power of 3 mW is applied.

**Figure 10 sensors-19-03561-f010:**
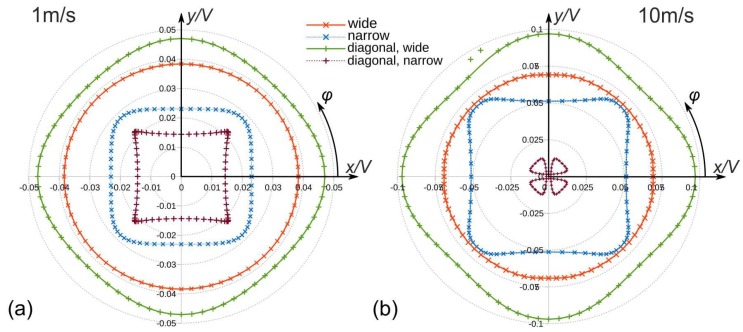
Computed directional diagrams of the bridge detuning signals for the bridge configurations according to Table 2. The marker positions correspond to 5-degree increments of the flow direction *φ*. A bridge supply voltage of 1 V and the values of the simulated thermistor resistances of Figure 8 are presumed. The shown characteristics belong to a mean cross-sectional flow velocity of (**a**) 1 m/s and (**b**) 10 m/s in the channel, respectively.

**Figure 11 sensors-19-03561-f011:**
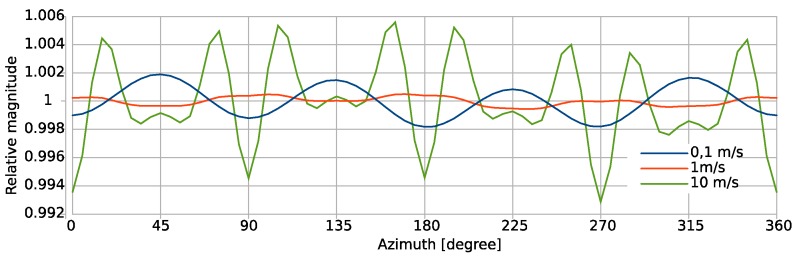
Relative magnitude variation of the voltage vectors generated by the “wide” thermistor bridge arrangement and mean flow velocities of 0.1, 1, and 10 m/s in the flow compartment of the finite element (FE) model.

**Figure 12 sensors-19-03561-f012:**
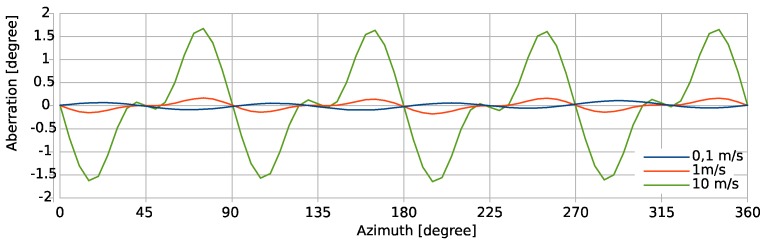
Simulated differences between the set angle of the flow and the angles computed from Equation (6) and the voltage vectors of the “wide” thermistor bridge arrangement.

**Figure 13 sensors-19-03561-f013:**
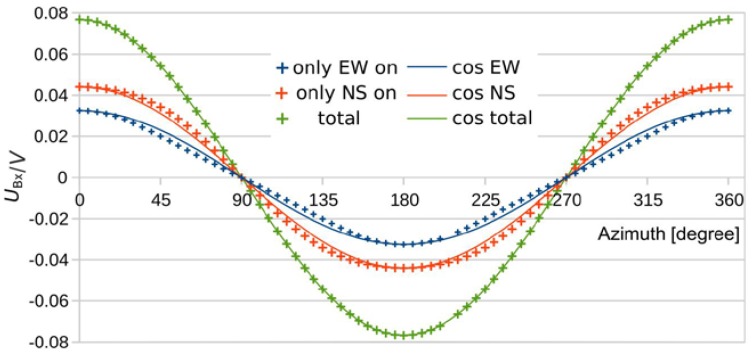
Partial and total response to 1 m/s flow as a function of the flow direction. The simulated characteristics are shown with respective cosine fits to highlight the deviations from these desired dependencies.

**Figure 14 sensors-19-03561-f014:**
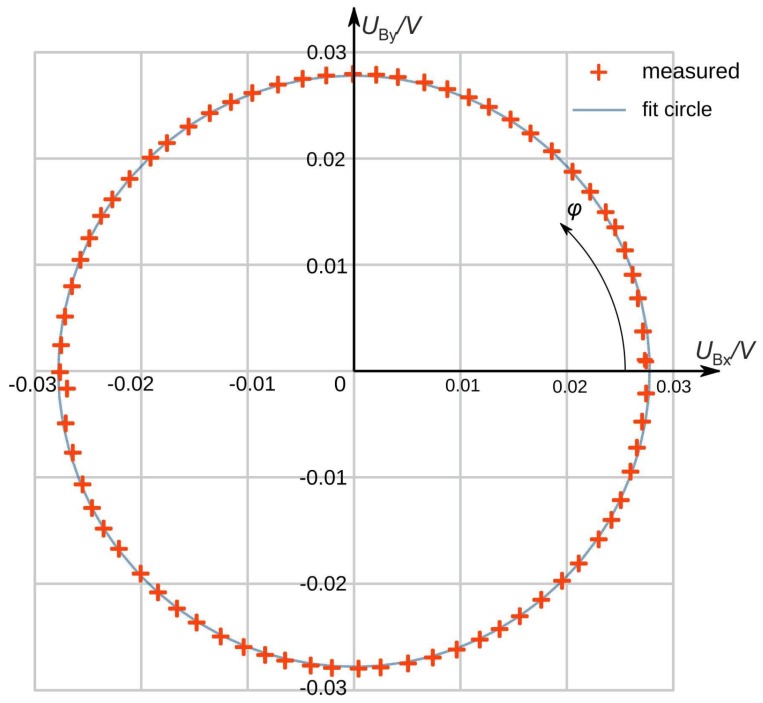
Measured directional diagram of a prototype of the advanced design aside with a circular fit. Zero-flow bridge offsets have been removed, the imposed average flow velocity was 1 m/s.

**Figure 15 sensors-19-03561-f015:**
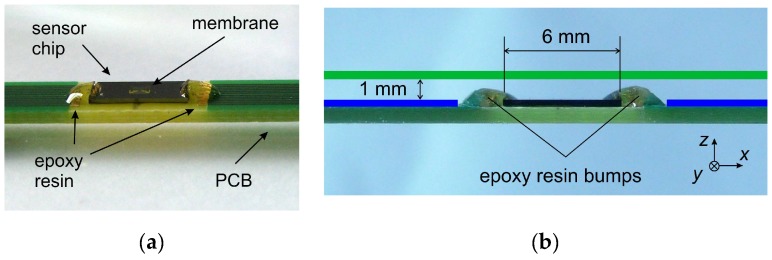
(**a**) Sensor chip (L × W × H: 6 × 3 × 0.35 mm^3^) mounted on the printed circuit board (PCB). The bonding wires are protected with epoxy resin. (**b**) Side view (orthogonal to *x–z*-plain). The bottom of the flow channel is indicated with the blue lines and its upper wall with the green line.

**Figure 16 sensors-19-03561-f016:**
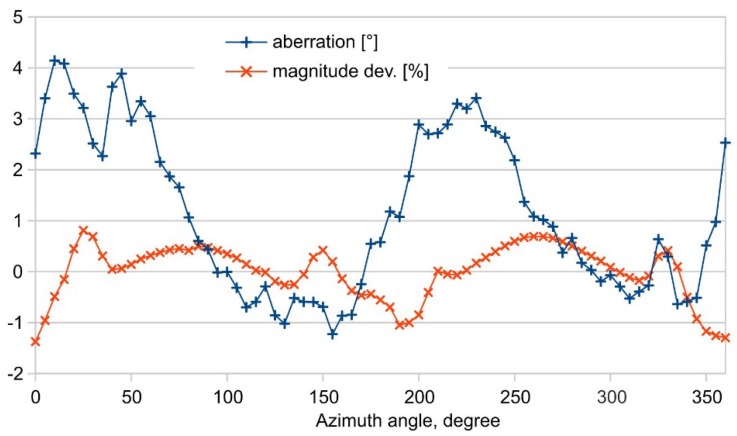
Measured relative magnitude variation (in percentage) and aberration (in degrees) related to the directional characteristic of Figure 14.

**Table 1 sensors-19-03561-t001:** Best performing membrane design/operating mode combinations. Maximum aberration is the largest difference between the flow direction set for simulation and the angle φ calculated using simulation results and Equation (4). Maximum magnitude variation is the difference between the velocity magnitude set for simulation and the calculated magnitude using Equation (4).

Design						
Error	Velocity	Triangular Calorimetric	Triangular, Self-Heating	Trapezoidal, Self-Heating	Circular, Self-Heating	Circular Calorimetric
max. aberration [degree]	1 m/s	0.17	0.4	0.16	0.4	0.8
	10 m/s	1.7	1.27	1.7	2.2	1.5
max. magnitude variation [%]	1 m/s	0.07	0.34	0.23	0.45	1.2
	10 m/s	0.5	0.52	3.5	4.2	3

**Table 2 sensors-19-03561-t002:** Distinct configurations of double resistance bridges.

Preferred Sensitivity Direction	±*x*	±*y*
circuit variant	main vs minor bridge diagonal	main vs minor bridge diagonal
wide	TH2, TH3 vs. TH6, TH7	TH1, TH8 vs. TH4, TH5
narrow	TH1, TH4 vs. TH5, TH8	TH2, TH7 vs. TH3, TH6
preferred sensitivity direction	45°, 225°	135°, 315°
diagonal wide	TH1, TH2 vs. TH5, TH6	TH7, TH8 vs. TH3, TH4
diagonal narrow	TH3, TH8 vs. TH4, TH7	TH1, TH6 vs. TH2, TH5

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
