# Peer review of "FEM-Analysis of 2D Micromachined Flow Transduers based on aGe-Thermistor Arrays and a Double Bridge Readout"

_sensors, 2019, doi:10.3390/s19163561_

Round 1

Reviewer 1 Report

The manuscript present a solid contribution and a systematic study at simulation level supported through experimental results.

Explanations are not always clear and should be included, please find the details in the scanned copy with handwritten annotations.

Author Response

Dear Reviewer,

We would like to thank the referee for many valuable suggestions and remarks in order to help us to improve the quality and readability of our paper. We have done our best to incorporate all recommendations as clearly as possible.

All changes in the latest manuscript version are marked by using a red color.

Page 2:

- Line 48 in Section “1. Introduction”: „iiii“ was changed to „iv“.

- Line 63 in Section “1. Introduction”: "[12]" was added.

Page 3:

- Line 83 in Section “1. Introduction”: “present some impressive simulation results” was changed to “present simulation results”.

- Line 96 in Section “2.1 Design Considerations”: “couple of system decision” was changed to “couple of system decision at system level”.

- Line 101 in Section “2.1 Design Considerations”: “are preferable” was changed to “are a common choice”.

- Figure 1 caption in Section “2.1 Design Considerations”: “This particular design and Wheatstone bridge configuration aims at pronounced ± vsensitivity” was changed to “This particular design (a) and Wheatstone bridge configuration (b) aims at pronounced sensitivity for flows along the the y-coordinate.”.

Page 4:

- Line 122 in Section “2.1 Design Considerations”: “Figure 1(a)” was added.

- Line 134, 135 in Section “2.1 Design Considerations”: after ” UBy,“ was added „as the four resistance values are approximately equal.“.

- Line 139, 140, 141 in Section “2.1 Design Considerations”: “Thus, putting the right thermistors in one bridge diagonal and the left thermistors in the other one provides preferential transduction of the x component of the velocity while neglecting its y component” was changed to 

“The best transduction of the x-component of the velocity can be achieved by swapping the position of RA and RD in Figure 1(b)

Page 5:

- Figure 2 in Section “2.1 Design Considerations”: 

“x”-axis was changed to “x/V”-axis” and

“y”-axis was changed to “y/V”-axis

Page 6:

- Line 175, 176, 177 in Section “2.1 Design Considerations”: “where the indicies 1 and 2 distinguish the devices and US denotes the preferred orientation (see Figure 2).” was changed to "where the indices 1 and 2 distinguish the devices and UB is the bridge signal for the preferred orientation of each device (i.e. UBy in Figure 2). Function f denotes the magnitude dependence for the preferred orientation (i.e. fy in Equation 3)”.

- Line 189, 190, 191 in Section “2.1 Design Considerations”: “The azimuthal uniformity of the flow magnitude conversion as well as the deviation between the flow direction and the estimated azimuth angle were considered as key parameters to validate the transducer performances.” was changed to “The key parameters of 2-D-transducer performance are azimuthal uniformity of the flow magnitude conversion as well as the deviation between the flow direction and the estimated azimuth angle aberration.

- Line 196-205: I added the Figure 3 in Section “2.2 FEM Modeling”: 

(b)

Figure 3: (a) Schematic cross section in y-direction of the flow sensor depicted in Figure 1 (m = 137.5 µm, n = 125 µm). (b) 3-D FEM model of the flow sensor. It considers only a narrow cylindrical section around the sensor membrane and the flow channel above. The colors illustrate the temperature distribution at the cross section through a membrane and parallel to the x-y-plane.  For this particular simulation a rectangular flow channel of 0.5 mm height and 1 mm width was assumed, with an airflow in y-direction and 1 m/s mean flow velocity. The heating power amounts to 3 mW and the ambient temperature to 25 °C (which is also the fluid temperature).

Page 9:

- Line 293-297: “Maximum aberration is the largest difference between the flow direction set for simulation and the angle ϕ calculated using simulation results and Equation 4. Maximum magnitude variation is the difference between the velocity magnitude set for simulation and the calculated magnitude using Equation 4.“ was added in Section “2.2 FEM Modeling”.

Page 11:

- Line 351, 352 in Section “2.3 Best performing membrane design”: “Right: Corresponding contour plot of the convection induced temperature change at the surface of transducer membrane.“ was changed to “Right: Corresponding contour plot of the convection induced temperature change in Kelvin at the surface of transducer membrane.“.

- Line 354, 355 in Section “2.4 Wheatstone bridge readout options”: “The eight thermistors of the design depicted in enable four distinct connection schemes for orthogonal pairs of Wheatstone bridges.“ was changed to „The eight thermistors of the design depicted in Figure 7 (confer also Figure 5(a), where the thermistors are denoted as R1R8) enable four distinct connection schemes for orthogonal pairs of Wheatstone bridges.“.

Page 12:

- Linie 384, 385 in Section “2.4 Wheatstone bridge readout options”: “Figure 6: Computed directional diagrams for the bridge configurations according to Table 2 and (3)” was changed to “Figure 10: Computed directional diagrams of the bridge detuning signals for the bridge configurations according to Table 2.“.

Page 15:

- Line 464-468 in Section “3. Measurements”: I added the paragraph: “The sensor chip is mounted flush with the bottom of a rectangular flow channel of 12 mm width and 1 mm height. The channel axes can be rotated 360° around the normal to the center of the membrane surface. Filtered nitrogen is used for the gas flow. The velocity of the nitrogen in the channel is adjusted by a mass flow controller with a range up to 2000 cm3/min, which corresponds to a maximum velocity of 27 m/s. The flow guidance setup has been described in detail in [20, 21].

Page 17:

- Line 492-493 in Section “3. Measurements”: “Figure 11: Measured relative magnitude variation [%] and aberration [°] related to the above directional characteristic.” was changed to “Figure 16: Measured relative magnitude variation [in percentage] and aberration [in degrees] related to the directional characteristic of Figure 14.”.

- Line 513 in Section “4. Conclusions”: “Fortunately” was changed to “However”.

Page 19:

- Line 581-583: I added in Section “References”: 

12. Talić A., Ćerimović S., Kohl F., Beigelbeck R., Schalko J., Keplinger F., “FEM and Measurement Analysis for Flow Sensor Featuring Three Different Operating Modes”, Procedia Eurosensors XXIV, Volume 5, 2010, Pages 746-749.”.

Reviewer 2 Report

This manuscript entitled “TFEM-analysis of 2-D micromachined flow transducers based on aGe-thermistors arrays and a double bridge readout” investigates detailed numerical analysis of velocity and angle sensor using microheaters and thermistors. High azimuthal uniformity was achieved in the proposed design from 0.1 to 10 m/s. The simulation results is quite important for the improved flow sensor design, therefore this manuscript offers valuable contributions in transducer fields. However, several descriptions need be enhanced to improve readability of the paper.onsequently, I would like to recommend this manuscript to be published in Sensors with minor revision. Followings are my items to be addressed.

- In 2.1, there is too little description about the simulated geometry of the sensor. I feel that the description in the manuscript, triangular, trapezoidal or circular, is not enough to understand the concept of the simulation. At least strategy or concept should be addressed in the context. 

- In 2.2, simulation part need adequate information for the readers to follow. Geometry, properties should be described. Also, the authors need to indicate heat transfer coefficient around the transducers.

- In 3, more information about the actual transducer is strongly needed to understand the results. If possible, I would recommend the authors to show pictures of bumps around the sensor which would be a reason of deviation in the measurement data.

Author Response

Dear Reviewer,

We would like to thank the referee for many valuable suggestions and remarks in order to help us to improve the quality and readability of our paper. We have done our best to incorporate all recommendations as clearly as possible.

All changes in the latest manuscript version are marked by using a red color (please see the attachment).

Comment 1:

In 2.1, there is too little description about the simulated geometry of the sensor. I feel that the description in the manuscript, triangular, trapezoidal or circular, is not enough to understand the concept of the simulation. At least strategy or concept should be addressed in the context. 

Thank you for this hint. We have added the schematic of the cross section in y-direction, 3D FEM model and some explanatory sentences. They can be now found at the beginning of the Section 2.2 (lines 196 – 230).

Comment 2:

In 2.2, simulation part need adequate information for the readers to follow. Geometry, properties should be described. Also, the authors need to indicate heat transfer coefficient around the transducers.

Thank you for this hint. We have added membrane view of three design alternatives, schematic illustration of this transducer and some explanatory sentences. They can be now found at the Section 2.2 (lines 246 – 283).

Comment 3:

In 3, more information about the actual transducer is strongly needed to understand the results. If possible, I would recommend the authors to show pictures of bumps around the sensor which would be a reason of deviation in the measurement data.

Thank you for this hint. We have added pictures of bumps around the sensors and some explanatory sentences. They can be now found at the beginning of the Section 3 (lines 463 – 488).